# Antibiotic Combination to Effectively Postpone or Inhibit the In Vitro Induction and Selection of Levofloxacin-Resistant Mutants in *Elizabethkingia anophelis*

**DOI:** 10.3390/ijms25042215

**Published:** 2024-02-12

**Authors:** Ching-Chi Lee, Chung-Hsu Lai, Chih-Hui Yang, Yi-Han Huang, Jiun-Nong Lin

**Affiliations:** 1Clinical Medical Research Center, National Cheng Kung University Hospital, College of Medicine, National Cheng Kung University, Tainan 704, Taiwan; chichingbm85@gmail.com; 2Departments of Internal Medicine, National Cheng Kung University Hospital, College of Medicine, National Cheng Kung University, Tainan 704, Taiwan; 3School of Medicine, College of Medicine, I-Shou University, Kaohsiung 824, Taiwan; laich6363@isu.edu.tw (C.-H.L.); je091410show@hotmail.com (Y.-H.H.); 4Division of Infectious Diseases, Department of Internal Medicine, E-Da Hospital, I-Shou University, Kaohsiung 824, Taiwan; 5Department of Biological Science and Technology, Meiho University, Pingtung 912, Taiwan; x00003196@meiho.edu.tw; 6Department of Critical Care Medicine, E-Da Hospital, I-Shou University, Kaohsiung 824, Taiwan

**Keywords:** combination therapy, fluoroquinolone, *Elizabethkingia anophelis*, quinolone resistance-determining region

## Abstract

Fluoroquinolones are potentially active against *Elizabethkingia anophelis*. Rapidly increased minimum inhibitory concentrations (MICs) and emerging point mutations in the quinolone resistance-determining regions (QRDRs) following exposure to fluoroquinolones have been reported in *E. anophelis*. We aimed to investigate point mutations in QRDRs through exposure to levofloxacin (1 × MIC) combinations with different concentrations (0.5× and 1 × MIC) of minocycline, rifampin, cefoperazone/sulbactam, or sulfamethoxazole/trimethoprim in comparison with exposure to levofloxacin alone. Of the four *E. anophelis* isolates that were clinically collected, lower MICs of levofloxacin were disclosed in cycle 2 and 3 of induction and selection in all levofloxacin combination groups other than levofloxacin alone (all *p* = 0.04). Overall, no mutations were discovered in *parC* and *parE* throughout the multicycles inducted by levofloxacin and all its combinations. Regarding the vastly increased MICs, the second point mutations in *gyrA* and/or *gyrB* in one isolate (strain no. 1) occurred in cycle 2 following exposure to levofloxacin plus 0.5 × MIC minocycline, but they were delayed appearing in cycle 5 following exposure to levofloxacin plus 1 × MIC minocycline. Similarly, the second point mutation in *gyrA* and/or *gyrB* occurred in another isolate (strain no. 3) in cycle 4 following exposure to levofloxacin plus 0.5 × MIC sulfamethoxazole/trimethoprim, but no mutation following exposure to levofloxacin plus 1 × MIC sulfamethoxazole/trimethoprim was disclosed. In conclusion, the rapid selection of *E. anophelis* mutants with high MICs after levofloxacin exposure could be effectively delayed or postponed by antimicrobial combination with other in vitro active antibiotics.

## 1. Introduction

The genus *Elizabethkingia*, which originated from *Flavobacterium* and *Chryseobacterium*, is a genus of Gram-negative, obligate aerobic, non-spore-forming, and glucose-nonfermenting bacilli. It normally exists or colonizes in the environment of communities and hospitals [1]. Among *Elizabethkingia* species, *Elizabethkingia anophelis* has recently been identified as a crucial pathogen responsible for severe nosocomial infections with substantial morbidity and mortality, particularly in immunocompromised individuals [1,2,3]. Administration of appropriate antibiotics is a cornerstone in successfully treating infections and saving lives, particularly in critically ill patients [4]. However, clinicians usually face the lethal challenge of treating patients infected by *E. anophelis* worldwide, as it notoriously reveals in vitro resistance to various clinically administered antimicrobials, such as the majority of beta-lactams, beta-lactam/beta-lactamase inhibitor combinations, colistin, and aminoglycosides [1,5]. To improve patient outcomes, the development of novel agents active against *E. anophelis* and additional antibiotic combinations is essential and urgent. 

Fluoroquinolones are traditionally active against *E. anophelis*, but their susceptibilities vary greatly in different areas [3,5,6,7]. Fluoroquinolone resistance commonly develops through the following mechanisms: point mutations in the quinolone resistance-determining regions (QRDRs) of DNA gyrase (*GyrA* and *GyrB*) and topoisomerase IV (*ParC* and *ParE*), plasmid-mediated genes encoding proteins that interfere with quinolone–enzyme interactions or enhance efflux, and chromosome-mediated genes resulting in over-expression of efflux pumps or under-expression of porins [8]. Among these mechanisms, point mutations in QRDRs have been recognized as the principal process of fluoroquinolone resistance against *E. anophelis* [9]. Importantly, the alteration of amino acids and the occurrence of point mutations in QRDRs after exposure to fluoroquinolones have recently been evidenced to be extremely rapid [10]. To avoid emerging antimicrobial resistance, antimicrobial combinations for treating patients infected with *E. anophelis* have inevitably been considered a good alternative [11]. Accordingly, we aimed to investigate the difference in the changes in minimum inhibitory concentrations (MICs) of fluoroquinolones against *E. anophelis* and the occurrence of mutations in QRDRs following levofloxacin exposure, compared with those following exposure to levofloxacin plus another active antibiotic reported in the literature. 

## 2. Results

### 2.1. Clones and MICs in E. anophelis 

On the basis of the analysis via Pulsed-Field Gel Electrophoresis (PFGE) for 142 clinical isolates, a phylogenetic tree was constructed, revealing 27 distinct clones of *E. anophelis*. Strains no. 1, 2, 3, and 4 were found to belong to clones 18, 8, 26, and 14, as shown in Appendix A. Of the four *E. anophelis* isolates, the MIC ranges for levofloxacin, minocycline, rifampin, cefoperazone/sulbactam, and sulfamethoxazole/trimethoprim were 0.5–1.0, 0.25–0.5, 0.25–0.5, 16–256, and 2–16 mg/L, respectively.

### 2.2. Changes in Levofloxacin MICs during Induction Cycles

The changes in levofloxacin MICs in levofloxacin alone and levofloxacin plus different concentrations (0.5× and 1 × MIC) of minocycline, rifampin, cefoperazone/sulbactam, or sulfamethoxazole/trimethoprim were exhibited in each cycle (Figure 1). After exposure to levofloxacin, the levofloxacin MICs apparently increased by the induction cycle of levofloxacin alone, while levofloxacin MICs in cycle 2 were significantly higher than those in cycle 0. Notably, in cycle 2 and 3, the levofloxacin MICs in all levofloxacin combination groups (all *p* = 0.04) were lower than those in levofloxacin alone (Figure 1A,B). In cycle 4, lower levofloxacin MICs were exhibited in the groups of levofloxacin plus 1 × MIC minocycline (Figure 1A), levofloxacin plus 0.5× or 1 × MIC rifampin (Figure 1A), levofloxacin plus 0.5× or 1 × MIC cefoperazone/sulbactam (Figure 1B), and levofloxacin plus 0.5× or 1 × MIC sulfamethoxazole/trimethoprim (Figure 1B), compared with levofloxacin alone (all *p* = 0.02). In cycle 5, levofloxacin MICs in the groups of levofloxacin plus 1 × MIC minocycline (Figure 1A), levofloxacin plus 0.5× or 1 × MIC rifampin (Figure 1A), levofloxacin plus 0.5× or 1 × MIC cefoperazone/sulbactam (Figure 1B), and levofloxacin plus 0.5× or 1 × MIC sulfamethoxazole/trimethoprim (Figure 1B) were lower than those of levofloxacin alone (all *p* < 0.001). In cycle 6 and 7, lower levofloxacin MICs were disclosed in levofloxacin plus 1 × MIC minocycline (Figure 1A), levofloxacin plus 0.5× or 1 × MIC rifampin (Figure 1A), levofloxacin plus 0.5× or 1 × MIC cefoperazone/sulbactam (Figure 1B), and levofloxacin plus 1 × MIC sulfamethoxazole/trimethoprim (Figure 1B), compared with levofloxacin alone (all *p* < 0.001). The levofloxacin MICs for levofloxacin alone and its combinations in each induction cycle are detailed in Appendix A.

### 2.3. MIC Changes in the Combined Antibiotics in Induction Cycles

The alterations in the MICs of levofloxacin-combined antibiotics, in terms of minocycline, rifampin, cefoperazone/sulbactam, and sulfamethoxazole/trimethoprim, were exhibited in each cycle (Figure 2). Irrespective of whether the subjects were exposed to 0.5× or 1 × MIC, the MIC changes after cycle 1 were not significant in minocycline (Figure 2A), rifampin (Figure 2B), and cefoperazone/sulbactam (Figure 2C). However, following exposure to levofloxacin, MICs of sulfamethoxazole/trimethoprim (0.5× or 1 × MIC) were significantly higher in cycle 1 than in cycle 0 (Figure 2D). The MICs of combined antibiotics during the induction cycles are detailed in Appendix A.

### 2.4. Mutations in QRDRs during Induction Cycles

Overall, no mutations were discovered in *parC* and *parE* throughout the multicycle of induction and selection by levofloxacin plus any active antibiotic. No non-synonymous mutations in *gyrA* and *gyrB* were detected following exposure to levofloxacin plus 0.5× or 1 × MIC rifampin (Appendix A) or levofloxacin plus 0.5× or 1 × MIC cefoperazone/sulbactam (Appendix A). 

In cycle 0 of levofloxacin plus 0.5× or 1 × MIC minocycline, the first point mutation in *gyrA* and/or *gyrB* of QRDRs was disclosed in strain no. 1 (Table 1). Regarding vastly increased MICs, the second point mutations in *gyrA* (83 Arg [AGG]) and *gyrB* (470 Glu [GAG]) appeared in cycle 2 following exposure to levofloxacin plus 0.5 × MIC minocycline and were delayed occurring in cycle 5 (83 Ile [ATC]) following exposure to levofloxacin plus 1 × MIC minocycline. For strain no. 3 (Table 1), the first point mutations (81 Gly [GGT]) were disclosed in cycle 0 and the second (81 Asp [GAT]) in cycle 4 in GyrA (*gyrA*) following exposure to levofloxacin plus 0.5 × MIC minocycline; no mutations occurred following the induction of levofloxacin plus 1 × MIC minocycline.

Following exposure to levofloxacin plus 0.5× or 1 × MIC sulfamethoxazole/trimethoprim (Table 2), no mutation in *gyrA* and/or *gyrB* of QRDRs was discovered in the three strains (no. 1, 2, and 4). For strain no. 3, the first mutation in GyrA (*gyrA*) (87 Asp [GAT]) and GyrB (*gyrB*) (471 Glu [GAA]) appeared in cycle 0 after exposure to levofloxacin plus 0.5×MIC, and the MICs vastly increased to 128 mg/L with the second point mutation in GyrA (*gyrA*) (87 Asp [GAT]) and GyrB (*gyrB*) (471 Glu [GAA]), which occurred in cycle 4 following exposure to levofloxacin plus 0.5 × MIC sulfamethoxazole/trimethoprim. However, point mutations in *gyrA* and/or *gyrB* were not discovered following exposure to levofloxacin plus 1 × MIC sulfamethoxazole/trimethoprim.

## 3. Discussion

In the past, *E. anophelis* has shown frequent resistance to most beta-lactams and other antimicrobials commonly prescribed in clinical practice. Studies have indicated that levofloxacin has variable in vitro activity against *E. anophelis* cross areas, with susceptibilities ranging from 29% to 96% [3,7]. Because of its multidrug-resistant nature, levofloxacin could be reasonably considered a viable option for treating individuals with *E. anophelis* infections. However, the widespread use of fluoroquinolones has led to the emerging development of fluoroquinolone-resistant microorganisms, posing a critical dilemma for global public health [8]. Therefore, to optimize treatment for *E. anophelis* infections, it is essential to understand the strategy to downregulate the development of resistant mutants and to prevent the widespread emergence of fluoroquinolone resistance.

With antibiotic resistance on the rise and a decline in the development of novel antibiotics, one of the most pivotal challenges is to preserve the effectiveness of the antibiotics currently available. The preferred method involves using two in vitro active antibiotics in combination because antimicrobial combination offers numerous advantages over monotherapy in terms of appropriate coverage as well as empirical administration, synergistic effect, and prevention of the emerging antimicrobial resistance [11]. In the literature, fluoroquinolones, minocycline, rifampin, cefoperazone/sulbactam, and sulfamethoxazole/trimethoprim are among the primary agents currently available for the treatment of individuals infected by *E. anophelis* [12,13,14]. This is the rationale behind selecting these antibiotics as potential candidates for combination with levofloxacin in the present study. 

Furthermore, the synergistic effect of antimicrobial combinations against *E. anophelis* in vitro, such as minocycline plus rifampin [13] and minocycline plus levofloxacin [12], has been reported in numerous studies. The prognostic advantage of piperacillin/tazobactam and fluoroquinolones or trimethoprim/sulfamethoxazole in combination therapy has been evidenced for *E. anophelis* infections in an observational study [15]. Hence, combination therapy for patients with *Elizabethkingia* infections has inevitably been considered a good alternative for clinicians [15,16]. In contrast to previous reports, our novel findings indicate that combining fluoroquinolone with other active agents, such as minocycline, rifampin, and sulfamethoxazole/trimethoprim, demonstrated benefits in preventing or delaying the emergence of fluoroquinolone resistance during fluoroquinolone therapy. Given the importance of fluoroquinolones in treating patients with *Elizabethkingia* infections, we believe that our principal finding holds a significant value for clinical practice in managing patients with *E. anophelis* infections.

The acquisition of point mutations in the QRDRs has been studied through in vitro exposure to increasing MICs of fluoroquinolones in numerous gram-positive microorganisms [17,18] and *E. anophelis* [10]. Numerous studies have illustrated that amino acid alternations of Ser83Ile and Ser83Arg in the QRDR of *gyrA* were the most common sites of mutations associated with fluoroquinolone resistance in *E. anophelis* [7,9]. Consistent with these reports, the present study revealed that Ser83Arg and Ser83Ile in GyrA occurred in one of the four isolates during the induction cycle of levofloxacin plus minocycline. Notably, the appearance of the second point mutations, resulting in significantly elevated MICs, was delayed after administering levofloxacin along with 1 × MIC minocycline, which was superior to the combination with 0.5 × MIC minocycline. In addition, point mutations in the QRDR manifest exclusively when levofloxacin is combined with 0.5 × MIC trimethoprim/sulfamethoxazole, not when it is combined with 1 × MIC trimethoprim/sulfamethoxazole. Accordingly, the results of this study provide valuable information concerning the appropriate dosage when using levofloxacin with another active agent to at least achieve MICs of the combined antibiotics in the infected tissue. 

This study possesses several limitations. First, despite testing four isolates belonging to different clones in the present study, the genetic linkage of these *E. anophelis* isolates with worldwide and Taiwan strains, as recognized by core-genome multilocus sequence typing (cg-MLST) and whole-genome comparative analysis [19], was limited. Second, despite point mutations in QRDRs previously being recognized as the principal process of fluoroquinolone resistance against *E. anophelis* [9,10], the effect of porin loss and the contribution of efflux pumps on increasing MICs was not studied in the present study. Third, although the advantages of levofloxacin in combination with other active antimicrobials have been revealed herein, comprehensive information detailing the molecular mechanisms responsible for these benefits is scant. Therefore, an investigation to assess the clinical efficacy and/or mechanism of these combination therapies will be necessary in the near future.

## 4. Methods

### 4.1. Study Setting and Bacterial Isolates

Four clinical *E. anophelis* isolates (both levofloxacin and ciprofloxacin MICs ≤ 1 mg/L) were randomly selected from two hospitals in southern Taiwan, namely E-Da Hospital and E-Da Cancer Hospital. The Institutional Review Board of E-Da Hospital (EMRP-110-177) approved the study protocol in accordance with the principles of the Declaration of Helsinki and the national standards of Taiwan. Informed consent requirements were waived because of the retrospective analysis of clinical isolates. As previously mentioned [10], *E. anophelis* was confirmed through 16S rRNA gene sequencing prior to the induction and the resistant selection process. Because our previous findings indicate the absence of mutations in the QRDRs in the isolates with levofloxacin MICs ≤ 32 mg/L during the in vitro induction and selection of levofloxacin-resistant mutants [10], we further examined only those with MICs ≥ 64 mg/L to identify any mutation in QRDRs.

### 4.2. PFGE

DNA from *E. anophelis* isolates was digested using XbaI, and all isolates underwent PFGE analysis to determine their clonal relationship, as previously described [2]. In brief, the digested DNA fragments were separated using a contour-clamped Homogeneous Electric Field Mapper XA Chiller System (Bio-Rad, Hercules, CA, USA). A dendrogram was generated using GelCompar II software (version 6; Applied Maths, Sint-Martens-Latem, Belgium). Isolates were considered to have a clonal relationship if they shared a similarity of ≥80% between the patterns of their DNA fragments.

### 4.3. Determination of the Minimum Inhibitory Concentrations

The susceptibility of *E. anophelis* to levofloxacin (Sigma-Aldrich, St. Louis, MO, USA) was assessed by determining the MICs using 96-well broth microdilution panels, following the manufacturer’s guidelines (Thermo Fisher Scientific, Oakwood Village, OH, USA). The antibiotics that were candidates for combination with levofloxacin were minocycline, rifampin, cefoperazone/sulbactam, and sulfamethoxazole/trimethoprim, all obtained from Sigma-Aldrich, St. Louis, MO, USA. The interpretation of susceptibility was based on the breakpoints outlined in the 2022 Clinical and Laboratory Standards Institute guidelines [20] for “other non-Enterobacteriaceae”. Levofloxacin susceptibility was defined as an MIC of ≤2 mg/L, whereas MICs of 4 and ≥8 mg/L indicated intermediate sensitivity and resistance, respectively. 

### 4.4. Multicycle Induction and Selection of Resistant Mutants

The acquisition of point mutations in the QRDRs has been studied through in vitro exposure to fluoroquinolones in numerous microorganisms [17,18]. The parent strains were cultured on antibiotic-free Muller–Hinton (MH) agar plates (Becton, Dickinson and Company, Sparks, MD, USA) at 37 °C for 24 h. Each isolate was then inoculated in MH broth at a concentration of 2 mL of 1 × 10^8^ cfu/mL (approximately 0.5× McFarland standard) and exposed to the levofloxacin MIC at 37 °C for 24 h. Afterward, 200 µL of the inoculated broth was plated at 37 °C for 24 h on MH broth plates containing different concentrations of levofloxacin (0×, 0.5×, 1×, and 2 × MICs) alone and its combinations (all, 0.5×, and 1 × MIC) with minocycline, rifampin, cefoperazone/sulbactam, and sulfamethoxazole/trimethoprim. From the colonies on the MH broth plates containing the highest concentrations of levofloxacin, three colonies were randomly selected and subcultured to determine the MICs. The colonies that survived under the highest MIC were chosen to repeat the cycle of induction and mutant selection until the MIC reached >256 mg/L or for seven cycles. 

### 4.5. Amplification and Sequencing to Identify Mutations in QRDRs

The DNA sequence was analyzed to investigate mutations in the QRDRs of *gyrA*, *gyrB*, *parC*, and *parE*. The amplification of fragments was achieved using the primers previously established [10]. The DNA sequencing reaction was conducted using the Wizard Genomic DNA Purification Kit from Promega (Madison, WI, USA) and the Applied Biosystems 3730xl DNA Analyzer from Perkin-Elmer (Applied Biosystems, Foster City, CA, USA). 

### 4.6. Statistical Analyses

Statistical analyses were conducted using the Statistical Package for the Social Sciences for Windows (Version 23.0; Chicago, IL, USA). The statistical difference in the distribution of MICs was assessed using Student’s *t*-test. All *p*-values were examined using a two-tailed test, and a *p*-value of <0.05 was considered statistically significant. 

## 5. Conclusions

For *E. anophelis* susceptible to levofloxacin and ciprofloxacin, levofloxacin combination with another in vitro active antibiotic, such as minocycline, rifampin, cefoperazone/sulbactam, or sulfamethoxazole/trimethoprim, could be effective in delaying the rapid increase in levofloxacin MICs and inhibiting point mutants in QRDRs after exposure to levofloxacin. In addition, we demonstrated the optimal dosage of the antibiotic combination required to attain concentrations of the combined antibiotics (such as sulfamethoxazole/trimethoprim and minocycline) above or equal to the MICs in the infected tissue. Therefore, in addition to the judicious use of fluoroquinolones, administration along with another active antibiotic might be considered to avoid the rapid emergence of fluoroquinolone resistance and treatment failure in patients infected by *E. anophelis.*

## Figures and Tables

**Figure 1 ijms-25-02215-f001:**
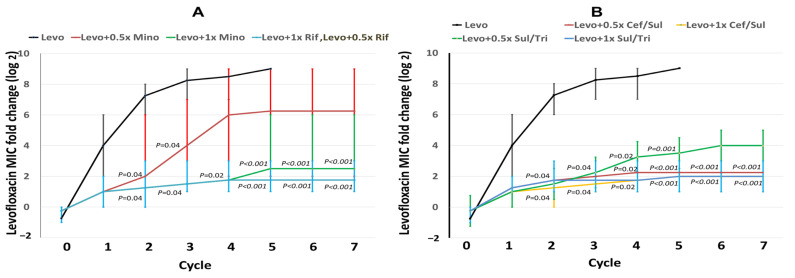
Fold changes in fluoroquinolone minimum inhibitory concentrations (MICs) in selected *E. anophelis* mutants. Four wild-type isolates were used as parent strains to be exposed to levofloxacin or its combination with minocycline (**A**), rifampin (**A**), cefoperazone/sulbactam (**B**), and sulfamethoxazole/trimethoprim (**B**) in a stepwise manner. The *x*-axis numbers represent the cycles of induction and selection. The *y*-axis numbers indicate the average (lines) and range (vertical bars) of MIC fold changes (log_2_) in each cycle. The *p* value indicates the difference in MIC distributions between exposure to levofloxacin alone and the levofloxacin combination within the same cycle.

**Figure 2 ijms-25-02215-f002:**
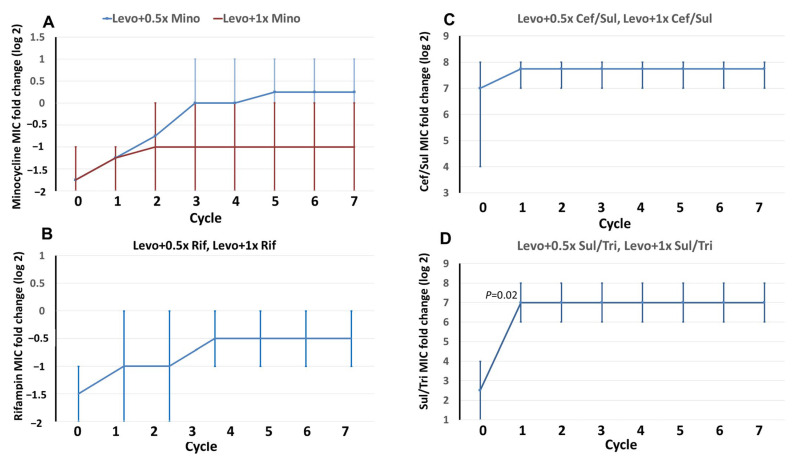
Fold changes in minimum inhibitory concentrations (MICs) in selected mutants of *E. anophelis*. Four wild-type isolates were used as parent strains to be exposed to levofloxacin in combination with minocycline (**A**), rifampin (**B**), cefoperazone/sulbactam (**C**), and sulfamethoxazole/trimethoprim (**D**) in a stepwise manner. The *x*-axis numbers represent the cycle of induction and selection. The *y*-axis numbers indicate the average (lines) and range (vertical bars) of the MIC fold changes (log_2_) in each cycle. The *p*-value demonstrates the difference in MIC distributions between cycle 0 and the indicated cycle. In Figure 2B–D, one curve represents two different concentrations of combined antimicrobials.

**Table 1 ijms-25-02215-t001:** Minimum inhibitory concentrations (MICs) of levofloxacin against *E. anophelis* and point mutations in quinolone resistance-determining regions in each step of multicycle induction and mutant selection by levofloxacin plus minocycline (0.5× and 1×MIC).

Levofloxacin + 1 × MIC Minocycline					1 × Levofloxacin + 0.5 × MIC Minocycline				
Strain	Cycle	Levofloxacin MIC (mg/L)	Point Mutation in Gyr A (*gyrA*)	Point Mutation in GyrB (*gyrB*)	Strain	Cycle	Levofloxacin MIC (mg/L)	Point Mutation in Gyr A (*gyrA*)	Point Mutation in GyrB (*gyrB*)
No. 1	0	0.5	83 Ser (AGC)	N/D	No. 1	0	0.5	83 Ser (AGC)	470 Glu (GAG)
	1	4	N/D	N/D		1	4	N/D	N/D
	2	8	N/D	N/D		2	64	83 Arg (AGG)	470 Glu (GAG)
	3	8	N/D	N/D		3	128	83 Arg (AGG)	470 Glu (GAG)
	4	8	N/D	N/D		4	>256	83 Arg (AGG)	470 Lys (AAG)
	5	64	83 Ile (ATC)	N/D		-	-	-	-
	6	64	83 Ile (ATC)	N/D		-	-	-	-
	7	64	83 Ile (ATC)	N/D		-	-	-	-
No. 2	0	1	N/D	N/D	No. 2	0	1	N/D	N/D
	1	2	N/D	N/D		1	2	N/D	N/D
	2	2	N/D	N/D		2	2	N/D	N/D
	3	2	N/D	N/D		3	2	N/D	N/D
	4	4	N/D	N/D		4	4	N/D	N/D
	5	4	N/D	N/D		5	4	N/D	N/D
	6	4	N/D	N/D		6	4	N/D	N/D
	7	4	N/D	N/D		7	4	N/D	N/D
No. 3	0	1	N/D	N/D	No. 3	0	1	81 Gly (GGT)	N/D
	1	2	N/D	N/D		1	2	N/D	N/D
	2	2	N/D	N/D		2	2	N/D	N/D
	3	2	N/D	N/D		3	16	N/D	N/D
	4	2	N/D	N/D		4	>256	81 Asp (GAT)	N/D
	5	2	N/D	N/D		-	-	-	-
	6	2	N/D	N/D		-	-	-	-
	7	2	N/D	N/D		-	-	-	-
No. 4	0	1	N/D	N/D	No. 4	0	1	N/D	N/D
	1	1	N/D	N/D		1	1	N/D	N/D
	2	1	N/D	N/D		2	1	N/D	N/D
	3	2	N/D	N/D		3	16	N/D	N/D
	4	2	N/D	N/D		4	16	N/D	N/D
	5	2	N/D	N/D		5	32	N/D	N/D
	6	2	N/D	N/D		6	32	N/D	N/D
	7	2	N/D	N/D		7	32	N/D	N/D

ND = not detected.

**Table 2 ijms-25-02215-t002:** Minimum inhibitory concentrations (MICs) of levofloxacin against *E. anophelis* and point mutations in quinolone resistance-determining regions in each step of multicycle induction and mutant selection by levofloxacin plus sulfamethoxazole/trimethoprim (0.5× and 1 × MIC).

Levofloxacin + 1 × MIC Sulfamethoxazole/Trimethoprim					Levofloxacin + 0.5 × MIC Sulfamethoxazole/Trimethoprim				
Strain	Cycle	Levofloxacin MIC (mg/L)	Point Mutation in Gyr A (*gyrA*)	Point Mutation in GyrB (*gyrB*)	Strain	Cycle	Levofloxacin MIC (mg/L)	Point Mutation in Gyr A (*gyrA*)	Point Mutation in GyrB (*gyrB*)
No. 1	0	0.5	N/D	N/D	No. 1	0	0.5	N/D	N/D
	1	2	N/D	N/D		1	2	N/D	N/D
	2	8	N/D	N/D		2	8	N/D	N/D
	3	8	N/D	N/D		3	8	N/D	N/D
	4	8	N/D	N/D		4	8	N/D	N/D
	5	8	N/D	N/D		5	8	N/D	N/D
	6	8	N/D	N/D		6	8	N/D	N/D
	7	8	N/D	N/D		7	8	N/D	N/D
No. 2	0	1	N/D	N/D	No. 2	0	1	N/D	N/D
	1	4	N/D	N/D		1	4	N/D	N/D
	2	4	N/D	N/D		2	4	N/D	N/D
	3	4	N/D	N/D		3	4	N/D	N/D
	4	4	N/D	N/D		4	4	N/D	N/D
	5	8	N/D	N/D		5	8	N/D	N/D
	6	8	N/D	N/D		6	8	N/D	N/D
	7	8	N/D	N/D		7	8	N/D	N/D
No. 3	0	1	N/D	N/D	No. 3	0	1	87 Asp (GAT)	471 Glu (GAA)
	1	2	N/D	N/D		1	2	N/D	N/D
	2	2	N/D	N/D		2	2	N/D	N/D
	3	2	N/D	N/D		3	8	N/D	N/D
	4	2	N/D	N/D		4	128	87 Asp (GAT)	471 Lys (AAA)
	5	2	N/D	N/D		5	128	87 Asp (GAT)	471 Lys (AAA)
	6	2	N/D	N/D		6	>256	87 Asn (AAT)	471 Lys (AAA)
	7	2	N/D	N/D		-	-	-	-
No. 4	0	1	N/D	N/D	No. 4	0	1	N/D	N/D
	1	1	N/D	N/D		1	1	N/D	N/D
	2	1	N/D	N/D		2	1	N/D	N/D
	3	2	N/D	N/D		3	2	N/D	N/D
	4	2	N/D	N/D		4	2	N/D	N/D
	5	2	N/D	N/D		5	2	N/D	N/D
	6	2	N/D	N/D		6	2	N/D	N/D
	7	2	N/D	N/D		7	2	N/D	N/D

ND = not detected.

## Data Availability

Available from the corresponding author on reasonable request.

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
