# Peer review of "Antibiotic Combination to Effectively Postpone or Inhibit the In Vitro Induction and Selection of Levofloxacin-Resistant Mutants in Elizabethkingia anophelis"

_ijms, 2024, doi:10.3390/ijms25042215_

Round 1
Reviewer 1 Report
Comments and Suggestions for Authors
In this paper, the authors examined the combination effects of antibiotic levofloxacin and some other types of antibiotics on the resistant mutants in Elizabethkingia anophelis. Based on that, the authors concluded that the co-treatments could delay the increase of minimum inhibitory concentration during the antibiotic exposure. I have some comments and questions for the paper.
Line 71, full name of “PFGE” is needed for the 1st time shown in the manuscript.
Line 74, is there any specific reason why the authors chose “minocycline, rifampicin…trimethoprim” as the co-treatment in the assays? Or could the authors give a brief explanation for the selection of these antibiotics in the assays?
In figure 1, it looks like that the data of levofloxacin only treatment (black curves) is missing after Cycle 5 in fig 1A and 1B.
What’s the number (N) of biological replicates in figure 1 and 2?
In figure 1 and 2, the authors showed the fold change of antibiotic MIC after each cycle with different treatments. It would be interesting and important to examine the bacterial survival rate after each cycle, which could be a great complement related to the change of MIC.
In figure 2, not sure why only one curve exists with two different concentration treatments in fig 2B, 2C and 2D.
In table 1, full name of “N/D” is needed in the table legend.
The authors focus on the point mutations in the QRDRs of gyrA, gyrB, parC and parE, which leads to fluoroquinolone resistance. I’m curious whether the authors observed any other types of mutations in addition to point mutations during the antibiotic treatments, eg, deletion mutation or others?
Author Response
In this paper, the authors examined the combination effects of antibiotic levofloxacin and some other types of antibiotics on the resistant mutants in Elizabethkingia anophelis. Based on that, the authors concluded that the co-treatments could delay the increase of minimum inhibitory concentration during the antibiotic exposure. I have some comments and questions for the paper.
Response: Greatly thanks for your substantial review and comments.
Line 71, full name of “PFGE” is needed for the 1st time shown in the manuscript.
Response: Thanks for your review. The full name had been inserted in line 72.
Line 74, is there any specific reason why the authors chose “minocycline, rifampicin…trimethoprim” as the co-treatment in the assays? Or could the authors give a brief explanation for the selection of these antibiotics in the assays?
Response: Many thanks for your question. In the introduction section (lines 65-69), we emphasize our aim, including comparing MIC changes and the occurrence of point mutations following exposure to levofloxacin alone versus levofloxacin in combination with another active antibiotic, as reported in the literature. Furthermore, in the second paragraph of the discussion section (lines 172-181), we further explain the reasons we chose these antimicrobials as the combined candidates.
In Figure 1, it looks like that the data of levofloxacin only treatment (black curves) is missing after Cycle 5 in fig 1A and 1B.
Response: Many thanks for your question. In our study design, we detailed the multicycle induction and selection of resistant mutants by levofloxacin or its combination with another active antimicrobial. This process continues until the MIC reaches >256 mg/L or for seven cycles. Please refer to Section 4.4, "Multicycle Induction and Selection of Resistant Mutants" in the method section (lines 266-268).
What’s the number (N) of biological replicates in figure 1 and 2?
Response: Many thanks for your question. In our study design, a total of four clinical isolates were chosen for analysis. Please refer to Section 4.1, " Study Setting and Bacterial Isolates" in the method section (lines 224-225).
In Figure 1 and 2, the authors showed the fold change of antibiotic MIC after each cycle with different treatments. It would be interesting and important to examine the bacterial survival rate after each cycle, which could be a great complement related to the change of MIC.
Response: Many thanks for your opinion and suggestion. In our study design, we detailed the multicycle induction and selection of resistant mutants by levofloxacin or its combination with another active antimicrobial. This process has been comprehensively established in previous investigations dealing with numerous microorganisms, such as Staphylococcus aureus (reference No. 18), Streptococcus pneumoniae (reference No. 17), and Elizabethkingia anophelis (reference No. 10). Please refer to lines 195-197 and 255-256.
In this process, the measures included the change in MICs and the occurrence of point mutations. Unlike animal models, a survival rate is not indicative because all colonies can survive during the multicycle
In Figure 2, not sure why only one curve exists with two different concentration treatments in fig 2B, 2C and 2D.
Response: Many thanks for your opinion and question.
In Figure 2B, 2C, and 2D, one curve represents two different concentrations of combined antimicrobials. To avoid confusion for the reader, the sentence describing these figures has been added. Please refer to line 125-126.
In table 1, full name of “N/D” is needed in the table legend.
Response: Many thanks for your review. The full name of “N/D” was inserted (line 156 and 159).
The authors focus on the point mutations in the QRDRs of gyrA, gyrB, parC and parE, which leads to fluoroquinolone resistance. I’m curious whether the authors observed any other types of mutations in addition to point mutations during the antibiotic treatments, eg, deletion mutation or others?
Response: Many thanks for your opinion and suggestion. As your opinion, we focused solely on point mutations in QRDRs in our study. The impact of porin loss and the role of efflux pumps in increasing MICs were not investigated. Therefore, this limitation is highlighted in the limitations section (line 213-216).

Reviewer 2 Report
Comments and Suggestions for Authors
The article describes that how an antibiotic combination can successfully delays or prevents the in vitro emergence and selection of levofloxacin-resistant mutants in Elizabethkingia anophelis. Some minor correction are required in English. Conclusion should be more informative so please include some results.
Comments on the Quality of English LanguageMinor editing of English language required
Author Response
The article describes that how an antibiotic combination can successfully delays or prevents the in vitro emergence and selection of levofloxacin-resistant mutants in Elizabethkingia anophelis. Some minor correction are required in English. Conclusion should be more informative so please include some results.
Response: Many thanks for your substantial review and comments. The language aspect of the entire paper has been reviewed for syntactical and grammatical errors by a native English speaker, and the editorial certification was attached below this letter. Furthermore, the conclusion section has been revised to emphasize our findings (line 282-291).

Reviewer 3 Report
Comments and Suggestions for Authors
This research focuses on a minor but important issue in healthcare, namely the emergence of antibiotic resistance in E. anophelis, a pathogenic microorganism that causes serious infections, especially in those with weakened immune systems. The research investigates the use of combination treatments as a possible strategy to postpone the development of resistance. This technique aims to prolong the effectiveness of existing antibiotics and is considered novel. The authors use a systematic step-by-step strategy to examine the impact of levofloxacin in conjunction with other antibiotics in E. anophelis. The research notes a postponement in the occurrence of resistance while using combination therapy, which has significance for the formulation of new treatment recommendations.
Nevertheless, the research is performed in a controlled laboratory setting, and its findings may not consistently reflect real-life outcomes. Bacterial activity might vary within the intricate surroundings of a human host. The study examines a limited number of antibiotic combinations. Expanding the scope of antibiotic testing might provide more complete and thorough information. Although the research demonstrates that combination treatment may postpone the development of resistance, it does not provide a comprehensive explanation of the underlying molecular processes responsible for this phenomenon. The study focuses on a restricted set of E. anophelis strains, perhaps limiting the representation of resistance development among various isolates. The study primarily examines mutations in the QRDR, potentially overlooking other resistance mechanisms and thus potentially underestimating the intricacy of resistance development.
Overall, the research is well-written and accomplished. Perhaps the small number of strains is the only reason to consider adopting the results. However, I should highlight that it is a less often isolated pathogen. The critiques I have given are thus more of a tip for future development than a job to be improved at now. I consequently suggest publishing it in its existing form.
Author Response
This research focuses on a minor but important issue in healthcare, namely the emergence of antibiotic resistance in E. anophelis, a pathogenic microorganism that causes serious infections, especially in those with weakened immune systems. The research investigates the use of combination treatments as a possible strategy to postpone the development of resistance. This technique aims to prolong the effectiveness of existing antibiotics and is considered novel. The authors use a systematic step-by-step strategy to examine the impact of levofloxacin in conjunction with other antibiotics in E. anophelis. The research notes a postponement in the occurrence of resistance while using combination therapy, which has significance for the formulation of new treatment recommendations.
Nevertheless, the research is performed in a controlled laboratory setting, and its findings may not consistently reflect real-life outcomes. Bacterial activity might vary within the intricate surroundings of a human host. The study examines a limited number of antibiotic combinations. Expanding the scope of antibiotic testing might provide more complete and thorough information. Although the research demonstrates that combination treatment may postpone the development of resistance, it does not provide a comprehensive explanation of the underlying molecular processes responsible for this phenomenon. The study focuses on a restricted set of E. anophelis strains, perhaps limiting the representation of resistance development among various isolates. The study primarily examines mutations in the QRDR, potentially overlooking other resistance mechanisms and thus potentially underestimating the intricacy of resistance development.
Response: Greatly thanks for your substantial review and comments. Numerous sentences belonged to the limitation paragraph had been inserted. First, the limitation detailing a comprehensive reason of the underlining molecular mechanism responsible the advantage of combinative therapy indicated in the present study was added (line 216-219). Moreover, because we focused solely on point mutations in QRDRs in our study, the impact of porin loss and the role of efflux pumps in increasing MICs were not investigated (line 213-216).
Overall, the research is well-written and accomplished. Perhaps the small number of strains is the only reason to consider adopting the results. However, I should highlight that it is a less often isolated pathogen. The critiques I have given are thus more of a tip for future development than a job to be improved at now. I consequently suggest publishing it in its existing form.
Response: Many thanks for your substantial review, again. In the limitation paragraph, we also suggest that further studies to assess the clinical efficacy and/or mechanisms of these combination therapies will be necessary in the near future (line 219-221).
Round 2
Reviewer 1 Report
Comments and Suggestions for Authors
Thank you for the responses. I'm Ok with the revision.